# Relative Handgrip Strength Mediates the Relationship between Hemoglobin and Health-Related Quality of Life in Older Korean Adults

**DOI:** 10.3390/healthcare10112215

**Published:** 2022-11-04

**Authors:** Jeonghyeon Kim, Inhwan Lee, Munku Song, Hyunsik Kang

**Affiliations:** College of Sport Science, Sungkyunkwan University, Suwon 16419, Korea

**Keywords:** quality of life, hemoglobin, muscle strength, chronic disease, gerontology

## Abstract

Background: Little is known regarding how anemia and handgrip strength influence the health-related quality of life (HRQoL) of older populations. This population-based study aimed to examine whether handgrip strength mediates the association between anemia and HRQoL in a representative sample of 6892 Korean adults aged ≥ 65 years (3753 females). Methods: HRQoL was assessed with the EuroQol-5 dimension. Anemia was defined according to World Health Organization (WHO) criteria, and handgrip strength was measured with a digital hand dynamometer. Results: Individuals with anemia were at increased risk for a lower HRQoL (odds ratio, OR = 1.285, *p* = 0.002) even after adjustment for age, sex, body mass index, income, marital status, education, physical activity, and multimorbidity compared with individuals without anemia (OR = 1). Individuals with weak handgrip strength were also at increased risk for a lower HRQoL (OR = 1.429, *p* < 0.001) even after adjustment for all the covariates compared with individuals with normal handgrip strength (OR = 1). Mediation analysis with a bootstrapping procedure showed that relative handgrip strength mediated the relationship between hemoglobin and HRQoL (95% confidence interval, CI 0.0192 to 0.0289) even after adjustment for all covariates, with 42.0% of the total effect of hemoglobin on HRQoL explained. Conclusions: The current findings suggest that the impact of anemia on HRQoL is partially mediated by weak handgrip strength, implying the clinical importance of having or maintaining adequate hemoglobin and handgrip strength via healthy lifestyle choices to maintain a high HRQoL later in life.

## 1. Introduction

An increase in life expectancy and a decrease in fertility rates have resulted in a larger proportion of the global population living longer [1]. Over the past few decades, South Korea has experienced the fastest growth of older adults in the total population among the OECD countries. South Korea was considered an aging society in 2000, an aged society in 2017, and will become a super-aged society by 2025 [2]. Considering this rapid population aging, developing non-pharmacologic strategies for healthy aging is important to maintain the stability and durability of the Korean healthcare system [3].

Normal aging is often accompanied by declines in overall health and physical, cognitive, and mental functioning (https://www.who.int/news-room/fact-sheets/detail/ageing-and-health accessed on 29 September 2022), collectively contributing to a decline in quality of life. Health-related quality of life (HRQoL) is a multidimensional concept that encompasses self-reported physical and mental health status over time [4]. Normal aging may lead to a decline in HRQoL due to its related adverse health outcomes [5], while health-promoting lifestyles positively impact HRQoL by reducing or attenuating adverse health outcomes associated with normal aging [6].

Anemia is common in older adults [7] and its prevalence increases with advanced age [8]. Anemia is an important surrogate for a lower level of HRQoL among elderly persons [9]. The negative impact of anemia on HRQoL has been reported in Korean populations [10] and ethnic agricultural workers living in rural areas of Thailand [11]. Additionally, anemia has a negative impact on HRQoL in patients with chronic kidney disease [12] or cancers [13]. 

Anemia is a condition of not having enough healthy red blood cells to carry oxygen to the body’s tissues. In active skeletal muscles, the reduced oxygen-carrying ability of blood due to anemia interferes with cell oxygenation and weakens muscles [14]. As a result, anemia has been utilized as a surrogate marker for health problems. For example, previous studies have shown that anemia is associated with lower grip strength in older adults [15] and reduced grip strength and increased mortality in older inpatient patients [16]. 

Sarcopenia is a chronic condition characterized by losses in muscle mass and strength and is also considered a surrogate for a lower level of HRQoL, especially in elderly persons. From this perspective, loss of muscle strength assessed by handgrip strength is associated with health conditions, sarcopenia, disability, impaired physical functioning, and cognitive decline in older adults [17]. Additionally, handgrip strength is a biomarker for a lower HRQoL in older populations [18,19,20] and cancer survivors [21,22].

Taken together, it is reasonable to assume that anemia may predict a decline in HRQoL due to low handgrip strength. However, few studies have investigated if anemia and handgrip strength predict HRQoL later in life. In the current study, we aimed to investigate (1) the associations between anemia and handgrip strength with HRQoL and (2) the impact of anemia on HRQoL through weak handgrip strength in a representative sample of older Korean adults.

## 2. Materials and Methods

### 2.1. Data Source

This study used the data extracted from the 2014-2019 Korea National Health and Nutrition Examination Survey (KNHNES VI–VIII), a nationwide survey examining the health status, health behaviors, and food and nutrient consumption of the Korean population. As shown in Figure 1, a total of 37,491 adults aged 19 years and older participated in the 2014–2019 KNHNES surveys. For the current study purpose, we excluded 27,007 participants aged 19~64 years and included 10,484 older adults (4532 men) aged 65 years and older only. Respondents with no information regarding HRQoL (*n* = 457), handgrip strength (*n* = 1557), health behaviors (*n* = 741), or covariates (*n* = 837) were excluded. Data from the remaining 6892 individuals (3139 males) were used in the final analyses. Detailed information regarding KNHNES is available through the national public database (https://www.cdc.gov/nchs/nhanes/index.htm accessed on 1 August 2022).

### 2.2. Measurements

#### 2.2.1. Health-Related Quality of Life

HRQoL was assessed using the EuroQol-5 dimension (EQ-5D). The EQ-5D instrument measures the quality of life in five dimensions, namely mobility, self-care, usual activities, pain/discomfort, and anxiety/depression, rated on three levels (no problems, some or moderate problems, or extreme problems) [23]. The EQ-5D index was calculated using the Korean valuation set [24], and the lowest quartile of the index was considered a lower HRQoL.

#### 2.2.2. Anemia

Serum hemoglobin was measured based on the SLS hemoglobin detection method with a Sysmex XN-9000 Automated Hematology Analyzer (Sysmex Corporation, Kobe, Japan), which uses cyanide-free sodium lauryl sulfate (SLS). Anemia was defined according to the WHO criteria of serum hemoglobin concentrations of less than 13.0 g/dL for men, less than 12.0 g/dL for non-pregnant women, and less than 11.0 g/dL for pregnant women [25]. 

#### 2.2.3. Handgrip Strength

Handgrip strength was measured with a digital hand dynamometer (digital grip strength dynamometer, T.K.K 5401, Takei Scientific Instruments Co., Ltd., Tokyo, Japan). Each participant in a standing position exerted maximal handgrip strength three times with the dominant hands with a 30 s rest interval between each measurement [26]. To minimize variation in maximum handgrip strength between the dominant and non-dominant hands [26], we averaged the three attempted values of the right and left hands and expressed them as a relative term (kg/body mass index).

#### 2.2.4. Covariates

Covariates measured in the current study included age (years), sex (male vs. female), body mass index (BMI), income (Korean won per month), marriage (married with a spouse vs. married without a spouse vs. never married), education (elementary or less vs. middle or high school vs. college or higher), heavy alcohol consumption, smoking status, physical activity, and multimorbidity. Covariates were assessed using a self-reported questionnaire [27]. Past smokers (at least 100 cigarettes in their lifetime) and current smokers were categorized as smokers, and those who never smoked in their lifetime were categorized as non-smokers. Heavy alcohol drinkers were defined as those who consumed >14 drinks per week for men and >7 drinks per week for women [28]. Physical activity data were collected using a Korean version of the Global Physical Activity Questionnaire (GPAQ). The amount of time that respondents participated in activities during the last 7 days was expressed in units of Metabolic Equivalent of one Task-minute (MET) per week. The validity and reliability of the GPAQ were previously confirmed in the Korean population [29]. Multimorbidity was defined as the coexistence of two or more chronic health conditions diagnosed by physicians [30].

### 2.3. Statistics

Prior to statistical analyses, the normality of data was checked using QQ-plots and histograms. All data are presented as means (standard deviations) and numbers of cases (percentages) for continuous and categorical variables, respectively. Student’s *t*-test and the chi-square were used to compare continuous and categorical variables, respectively. Bivariate logistic regression was used to estimate the odds ratios (ORs) and 95% confidence intervals (CIs) of a lower HRQoL by anemia status and levels of handgrip strength. A linear regression analysis was used to estimate the correlation coefficients of the determinants for HRQoL. Finally, the Process Macro by Andrew F. Hayes was used to conduct a mediation analysis of the effect of relative handgrip strength (M) on the association between hemoglobin (X) and HRQoL (Y), as shown in Figure 2. Bias-corrected bootstrapping (*n* = 10,000) and 95% confidence intervals were used to evaluate the statistical significance of the mediation model. Otherwise, statistical significance was evaluated using an *α*-value of 0.05 with SPSS-PC (version 27.0, IBM Corporation, Armonk, NY, USA).

## 3. Results

Descriptive statistics of study participants by gender are provided in Table 1. Older Korean women were less educated (*p* < 0.001), less physically active, more likely to live without a spouse (*p* < 0.001), and less likely to drink (*p* < 0.001) and smoke (*p* < 0.001) than older Korean men. Additionally, older Korean women had a lower BMI (*p* < 0.001), lower income (*p* < 0.001), higher rate of multimorbidity (*p* < 0.001), and higher EQ-5D problems (*p* < 0.001); however, they had lower hemoglobin (*p* < 0.001), lower absolute and relative handgrip strength (*p* < 0.001 and *p* < 0.001, respectively), and lower EQ-5D index (*p* < 0.001) than older Korean men.

ORs and 95% CIs of HRQoL by anemia status and levels of handgrip strength are presented in Table 2. Individuals with anemia were at increased risk for a lower HRQoL (OR = 1.727, *p* < 0.001) compared to individuals without anemia. The increased risk for a lower HRQoL remained statistically significant (OR = 1.285, *p* = 0.002) after adjustments for age, gender, income, marriage, education, smoking, alcohol intake, physical activity, and relative handgrip strength. Likewise, individuals with weak handgrip strength were at increased risk for a lower HRQoL (OR = 1.668, *p* < 0.001) compared with individuals with normal handgrip strength. The increased risk for a lower HRQoL remained statistically significant (OR = 1.429, *p* < 0.001) after adjustment for all covariates.

Table 3 shows the correlation coefficients of the determinants for HRQoL. HRQoL was inversely correlated with age (*p* < 0.001), living without a spouse (*p* < 0.001), smoking (*p* = 0.001), and multimorbidity (*p* < 0.001); it positively correlated with income (*p* = 0.002), education (*p* < 0.001), physical activity (*p* = 0.002), hemoglobin (*p* < 0.001), and relative handgrip strength (*p* < 0.001).

Table 4 shows the association between hemoglobin and HRQoL mediated by relative handgrip strength. The Process macro for mediation analysis showed that hemoglobin had a direct effect on HRQoL (β_c’_ = 0.0064, *p* < 0.001) and an indirect effect on HRQoL through its effect on relative handgrip strength (β_ab_ = 0.0114, *p* < 0.001). Hemoglobin was positively associated with relative handgrip strength (β_a_ = 0.1092, *p* < 0.001), which was positively associated with HRQoL (β_b_ = 01046, *p* < 0.001) in Model 1. Hemoglobin and relative handgrip strength remained significant determinants of HRQoL even after adjustment for all measured covariates in Model 2 (β_c’_ = 0.0049, *p* = 0.002 and β_b_ = 0.0682, *p* < 0.001, respectively). 

The mediating effect of relative handgrip strength on the association between hemoglobin and HRQoL was further tested using a bootstrapping procedure (Table 4 and Figure 3). The bootstrap procedure showed that the 95% bias-corrected confidence interval (95% CI 0.0102 to 0.0127) was non-zero, indicating that relative handgrip strength mediates the relationship, with 64.0% of the total effect of hemoglobin on HRQoL explained. The mediating effect of relative handgrip strength remained non-zero (95% CI 0.0192 to 0.0289) after adjustment for all the covariates, with 42.0% of the total effect explained. Furthermore, the direct effect of hemoglobin (Hb) on HRQoL in the presence of the mediator was also significant (β = 0.0064, 95% CI = 0.0039 to 0.0089, *p* < 0.001), indicating that RHGS partially mediates the relationship between Hb and HRQoL.

## 4. Discussion

This population-based study examined the associations between hemoglobin, handgrip strength, and HRQoL in 6982 older Korean adults aged 65 years and older and showed that anemia and relative handgrip strength were independent predictors of a lower HRQoL. To the best of our knowledge, this is the first study to report that the impact of anemia on HRQoL is partially mediated by weak relative handgrip strength, implying that having and maintaining both adequate hemoglobin and normal handgrip strength is critical for a higher HRQoL later in life.

The current findings of the study are consistent with findings from previous studies that investigated the associations between HRQoL, anemia, and handgrip strength in different populations. For example, Kim et al. [10], by analyzing data obtained from the 2008–2016 KNHNES, showed that anemia was associated with an increased risk for a lower HRQoL in Korean adults aged 19 years and older. In a population-based cohort study involving 138,670 participants aged 18–93 years in the Netherlands, Wouters et al. [9] showed that anemia was significantly associated with a lower HRQoL and a lower survival rate in individuals aged 60 years and older. In a cross-sectional study involving 468 agricultural workers of various ethnicities (average age of 49.6 ± 13.8 years) in Thai border communities, Boonyathee et al. [11] showed that low hemoglobin levels associated with existing diseases and unhealthy behaviors were inversely associated with HRQoL.

The current findings of the study are also in line with the findings from previous studies reporting the associations between handgrip strength and HRQoL in older adults. For example, Baek, Kim, and Kim [20], by analyzing nationwide survey data, showed that weak handgrip strength and handgrip strength asymmetry (defined as a handgrip strength of >10% stronger on either hand) were associated with impaired HRQoL in older Korean adults. Kang, Lim, and Park [19] analyzed the 2015 KNHNES data and demonstrated that low handgrip strength correlated with poor HRQoL—especially mobility and pain/discomfort variables of the EQ-5D—in Korean adults aged 20 years and older. In a prospective population-based follow-up study, Taekema et al. [17] showed that poor handgrip strength predicted impaired ADL and cognitive decline in 555 older adults aged 85 years and older in the Netherlands. Additionally, previous studies have reported that handgrip strength is an important determinant of HRQoL in cancer survivors in Korea [21] and Spain [22]. Taken together, the findings from the current and previous studies support the prognostic roles of anemia and weak handgrip strength in identifying poor HRQoL later in life.

This is the first study to report that the association between anemia and poor HRQoL is partially mediated by weak handgrip strength. The mediating effect of weak handgrip strength on the impact of anemia on HRQoL can be explained as follows: Anemia, which is the impaired oxygen-carrying capacity of blood into working skeletal muscles, causes cellular deoxygenation called hypoxia [31]. As a result of hypoxia, cellular ATP levels drop, cellular functions cannot be maintained, and cells will die, resulting in a loss of muscle strength [31]. In turn, weakened muscle strength contributes to problems in mobility and dependency and decreases normal activities and activities of daily living [32]. Those adverse mental and physical health outcomes will collectively impair HRQoL. In support of this explanation, Gi et al. [15] showed that individuals with weak handgrip strength had a higher risk of anemia, with a greater association observed in males and in older adults aged 65 years and older, by analyzing data from the 2013–2017 KNHNES involving 16,638 Korean adults aged 19 years and older. Anemia has been reported to impact handgrip strength in older Australian men [33], older rural South African populations [34], and Georgian centenarians [35]. Additionally, handgrip strength was found to be a significant determinant of HRQoL in middle-aged patients with osteoarthritis [36] and in male patients with chronic obstructive pulmonary disease [37]. Taken together, the findings from the current and previous studies again suggest the importance of maintaining adequate hemoglobin levels and handgrip strength for higher HRQoL later in life.

This study has two major strengths. First, we are the first to report that anemia and weak handgrip strength are associated with poor HRQoL; the association between anemia and HRQoL is partially mediated by weak handgrip strength in older Korean adults. Second, we used data obtained from nationwide and well-designed systematic surveys. However, this study also has limitations. First, the cross-sectional design of this study limits any cause-effect explanations regarding our findings. Second, the findings of the study should be confirmed in other ethnic populations to allow for generalization. Third, we cannot completely rule out the chance of measurement errors in the self-reported covariate questionnaire.

## 5. Conclusions

Our study findings show that anemia may contribute to a lower HRQoL through weak handgrip strength, suggesting that promoting muscular strength via an active and healthy lifestyle is critical for better HRQoL in older Korean adults aged 65 years and older. 

## Figures and Tables

**Figure 1 healthcare-10-02215-f001:**
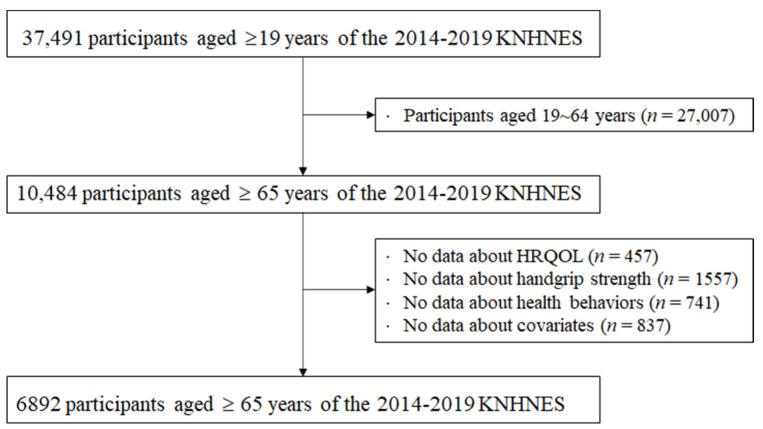
Flowchart for the selection of study participants. KNHNES: Korea National. Health and Nutritional Examination Survey; HRQOL: health-related quality of life.

**Figure 2 healthcare-10-02215-f002:**
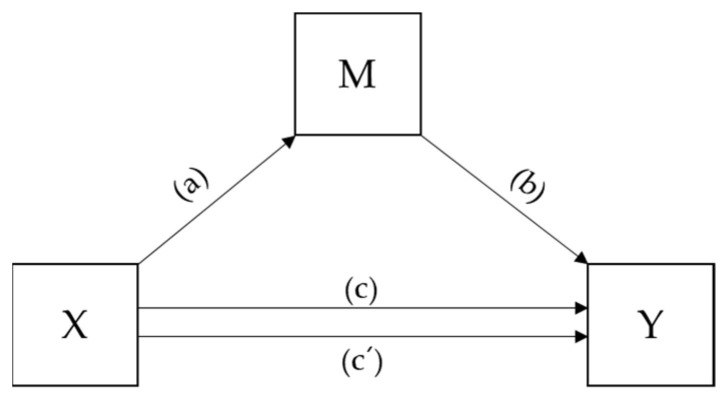
A theoretical model linking a cause (X) to an effect (Y) through a mediator (M). c = c’ + ab; c = total effect of X on Y; c´ = direct effect of X on Y; ab = indirect effect of X on Y through M.

**Figure 3 healthcare-10-02215-f003:**
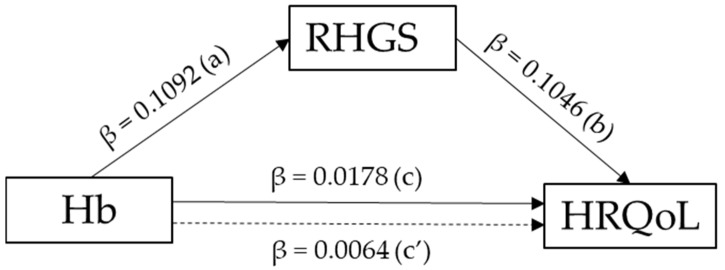
Mediation analysis. Path coefficients of hemoglobin (Hb) on health-related quality of life (HRQOL) through relative handgrip strength (RHGS). a: estimated coefficient for the regression with anemia predicting RHGS; b: estimated coefficient for the regression with RHGS predicting HRQoL; c = total effect of anemia on HRQoL; c´ = direct effect of anemia on HRQoL; ab = indirect effect of anemia on HRQoL through RHGS.

**Table 1 healthcare-10-02215-t001:** Descriptive statistics of study participants by gender.

Variables	Men (*n* = 3139)	Women (*n* = 3753)	Total (*n* = 6892)	*p*-Value
Age (year)	71.8 ± 5.2	71.9 ± 5.2	71.8 ± 5.2	0.582
BMI (kg/m^2^)	23.8 ± 2.9	24.5 ± 3.3	24.2 ± 3.1	<0.001
Income (10,000 won/month)	255.8 ± 263.0	218.8 ± 251.0	235.7 ± 257.2	<0.001
Education, n (%)				<0.001
Elementary school or less	1194 (38.0)	2603 (69.4)	3797 (55.1)	
Middle/high school	1405 (44.8)	956 (25.5)	2361 (34.3)	
College or higher	540 (17.2)	194 (5.2)	734 (10.6)	
Marital status, n (%)				<0.001
Married with a spouse	2768 (88.2)	2024 (53.9)	4792 (69.5)	
Married without a spouse	349 (11.1)	1701 (45.3)	2050 (29.8)	
Never married	22 (0.7)	28 (0.7)	50 (0.7)	
Current/past smoking, n (%)	2478 (78.9)	181 (4.8)	2659 (38.6)	
Heavy alcohol, n (%)	390 (12.4)	77 (2.1)	457 (6.8)	
Physical activity (METs per week)	846.5 ± 1480.3	526.6 ± 929.0	672.3 ± 1222.0	
Multimorbidity, n (%)				<0.001
0	765 (24.4)	527 (14.0)	1292 (18.7)	
1	1031 (32.8)	955 (25.4)	1986 (28.8)	
≥2	1343 (42.8)	2271 (60.6)	3614 (52.5)	
EQ-5D problems				
Mobility, n (%)	786 (25.0)	1542 (41.1)	2328 (33.8)	<0.001
Self-care, n (%)	185 (5.9)	386 (10.3)	571 (8.3)	<0.001
Usual activities, n (%)	391 (12.5)	792 (21.1)	1183 (17.2)	<0.001
Pain/discomfort, n (%)	765 (24.4)	1588 (42.3)	2353 (34.1)	<0.001
Anxiety/depression, n (%)	311 (9.9)	640 (17.1)	951 (13.8)	<0.001
EQ-5D index	0.926 ± 0.12	0.869 ± 0.16	0.895 ± 0.15	<0.001
Hemoglobin (g/dL)	14.5 ± 1.4	13.1 ± 1.1	13.7 ± 1.4	<0.001
AHGS (kg)	31.0 ± 6.8	18.4 ± 4.6	24.1 ± 8.5	<0.001
RHGS (kg/BMI)	1.31 ± 0.3	0.76 ± 0.2	1.01 ± 0.4	<0.001

AHGS: absolute handgrip strength; RHGS: relative handgrip strength; BMI: body mass index; EQ: EuroQol Group.

**Table 2 healthcare-10-02215-t002:** Odds ratios (ORs) and 95% confidence intervals (CIs) of a lower HRQoL by anemia and handgrip strength.

Predictors	Model 1		Model 2	
	OR (95% CI)	*p*-Value	OR (95% CI)	*p*-Value
Anemia				
Without anemia	1 (reference)		1 (reference)	
With anemia	1.727 (1.490~2.002)	<0.001	1.285 (1.096~1.507)	0.002
RHGS				
Normal	1 (reference)		1 (reference)	
Weak	1.668 (1.497~1.858)	<0.001	1.429 (1.274~1.604)	<0.001

LBMS: relative handgrip strength; HRQoL: health-related quality of life. Model 1: unadjusted. Model 2: adjusted for age, gender, income, marriage, education, smoking, alcohol intake, physical activity, morbidity, relative handgrip strength (for anemia), and hemoglobin (for relative handgrip strength).

**Table 3 healthcare-10-02215-t003:** Regression analyses for predictors of health-related quality of life.

Variables	β	SE	*p*-Value
Age	−0.03	0.036	<0.001
Body mass index	0.001	0.007	0.439
Sex	0.001	0.007	0.975
Income	0.001	0.001	0.002
Education	0.020	0.003	<0.001
Marital status	−0.014	0.004	<0.001
Smoking	−0.014	0.004	0.001
Heavy alcohol	0.003	0.007	0.635
Physical activity	0.001	0.001	0.002
Multimorbidity	−0.021	0.002	<0.001
Hemoglobin	0.005	0.000	<0.001
Relative handgrip strength	0.066	0.001	<0.001

**Table 4 healthcare-10-02215-t004:** The association between anemia and health-related quality of life in older Korean adults, mediated by relative handgrip strength.

Path	Model 1			Model 2		
	β (SE)	95% CI	*p*-Value	β (SE)	95% CI	*p*-Value
Anemia→RHGS, a	0.1092 (0.0028)	0.1037–0.1148	<0.001	0.0517 (0.0023)	0.0417–0.0563	<0.001
RHGS→HRQoL, b	0.1046 (0.0050)	0.0948–0.1145	<0.001	0.0682 (0.0066)	0.0553–0.0812	<0.001
Total effect, c	0.0178 (0.0012)	0.0154–0.0202	<0.001	0.0084 (0.0013)	0.0059–0.0110	<0.001
Direct effect, c’	0.0064 (0.0013)	0.0039–0.0089	<0.001	0.0049 (0.0013)	0.0023–0.0075	0.002
Indirect effect, ab	0.0114 (0.0006)	0.0102–0.0127		0.0035 (0.0025)	0.0192–0.0289	
Indirect to total effect (%)	64.0			42.0		

Model 1 was not adjusted. Model 2 was adjusted for age, body mass index, income, education, marital status, smoking, heavy alcohol, physical activity, and multimorbidity. The number of bootstrap samples for bias-corrected bootstrap confidence intervals is 10,000. RHGS: relative hand grip strength; HRQoL: health-related quality of life; CI: confidence interval; SE: standard errors. a: estimated coefficient for the regression with anemia predicting RHGS; b: estimated coefficient for the regression with RHGS predicting HRQoL; c = total effect of anemia on HRQoL; c´ = direct effect of anemia on HRQoL; ab = indirect effect of anemia on HRQoL through RHGS.

## Data Availability

The datasets used and analyzed during this study are available from the corresponding author upon reasonable request.

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
