# Peer review of "Relative Handgrip Strength Mediates the Relationship between Hemoglobin and Health-Related Quality of Life in Older Korean Adults"

_healthcare, 2022, doi:10.3390/healthcare10112215_

Round 1

Reviewer 1 Report

This study is expected to provide valuable information for maintaining the health of the elderly.

However, the following corrections and supplements are required.

1. Please supplement the suggestion of the need for research on the relationship between anemia and grip strength in introduction

2. A more detailed discussion of the causes that resulted in anemia, grip strength and quality of life is required.

3. 3. Please provide a clear rationale for why grip strength is considered a major variable in discussion.

Author Response

In our Response to Reviewer #1

We deeply appreciate the reviewers for their thoughtful comments. We did our best to address all the comments/critics point-by-point, which are highlighted in yellow color. Three references are newly added, and they are listed on the last page.

Q1) Please supplement the suggestion of the need for research on the relationship between anemia and grip strength in introduction

ANS1) Thanks. In our response to the comment, we revised the last three paragraphs in the Introduction as follows:

Anemia is a condition of not having enough healthy red blood cells to carry out oxygen to the body's tissues. In active skeletal muscles, the reduced oxygen-carrying ability of blood due to anemia interferes with cell oxygenation and weakens muscles [14]. As a result, anemia has been utilized as a surrogate marker for health problems. For example, previous studies have shown that anemia is associated with lower grip strength in older adults [15] and reduced grip strength and increased mortality in older inpatient patients [16].

Sarcopenia is a chronic condition characterized by losses in muscle mass and strength, and it is also considered a surrogate for a lower level of HRQoL, especially in elderly persons. From this perspective, loss of muscle strength assessed by handgrip strength is associated with health conditions, sarcopenia, disability, impaired physical functioning, and cognitive decline in older adults [17]. Additionally, handgrip strength is a biomarker for a lower HRQoL in older populations [18-20] and cancer survivors [21,22].

Taken together, it is reasonable to assume that anemia may predict a decline in HRQoL due to low handgrip strength. However, few studies have investigated if anemia and handgrip strength predict HRQoL later in life. In the current study, we aimed to investigate 1) the associations between anemia and handgrip strength with HRQoL and 2) the impact of anemia on HRQoL through weak handgrip strength in a representative sample of older Korean adults.

Q2) A more detailed discussion of the causes that resulted in anemia, grip strength and quality of life is required.

ANS2) Thanks for the comment. In our response to the comments, we provided the detailed discussion of the causes as follows:

“The mediating effect of weak handgrip strength on the impact of anemia on HRQoL can be explained as follows. Anemia, which is the impaired oxygen-carrying capacity of blood into working skeletal muscles, causes cellular deoxygenation called hypoxia [31]. As a result of hypoxia, cellular ATP levels drop, cellular functions cannot be maintained, and cells will die, resulting in a loss of muscle strength [31]. In turn, weakened muscle strength contributes to problems in mobility, decreases in normal activities and activities of daily living, and increased dependency [32]. Those adverse mental and physical health outcomes will collectively impair HRQoL. In support of this explanation, Gi et al. [15] showed that individuals with weak handgrip strength had a higher risk of anemia, with a greater association observed in males and in older adults aged 65 years and older, by analyzing data from the 2013-2017 KNHNES involving 16,638 Korean adults aged 19 years and older. Anemia has been reported to impact handgrip strength in older Australian men [33], older rural South African populations [34], and Georgian centenarians [35]. Additionally, handgrip strength was found to be a significant determinant of HRQoL in middle-aged patients with osteoarthritis [36] and in male patients with chronic obstructive pulmonary disease [37]. Taken together, the findings from the current and previous studies again suggest the importance of maintaining adequate hemoglobin levels and handgrip strength for higher HRQoL later in life”

ANS2) Considering the cross-sectional nature of the study, we don’t want to exaggerate in explaining the findings in a causal manner.

Q3) Please provide a clear rationale for why grip strength is considered a major variable in discussion.

ANS3) Weak handgrip strength with normal aging is attributable to various lifestyle risk factors, including anemia (Gi et al., 2020), physical inactivity (Seong et al., 2020), poor nutrition (Tak et al., 2016), and others (Seong et al., 2020).

List of Added References

  1. Puthucheary, Z.A.; Rawal, J.; McPhail, M.; Connolly, B.; Ratnayake, G.; Chan, P.; Hopkinson, N.S.; Phadke, R.; Dew, T.; Sidhu, P.S.; Velloso, C.; Seymour, J.; Agley, C.C.; Selby, A.; Limb, M.; Edwards, L.M.; Smith, K.; Rowlerson. A.; Rennie, M.J.; Moxham, J.; Harridge, S.D.; Hart. N.; Montgomery, H.E. Acute skeletal muscle wasting in critical illness. JAMA. 2013;310(15):1591-600. Erratum in: JAMA. 2014, 311, 625.
  2. Chang, S.Y.; Han, B.D.; Han, K.D.; Park, H.J.; Kang, S. Relation between handgrip strength and quality of life in patients with arthritis in Korea: the Korea National Health and Nutrition Examination Survey, 2015-2018. Medicina (Kaunas). 2022, 58, 172.
  3. Lee, S.H.; Kim, S.J.; Han, Y.; Ryu, Y.J.; Lee, J.H.; Chang, J.H. Hand grip strength and chronic obstructive pulmonary disease in Korea: an analysis in KNHANES VI. Int J Chron Obstruct Pulmon Dis. 2017, 12, 2313-2321.

Reviewer 2 Report

This paper presents an empirical study on relative handgrip strength
mediating the relationship between hemoglobin and health-related quality
of life in older Korean adults.

The manuscript is well written and the results of interest.

Important strengths include the huge sample size (approx. 10k
individuals) and the detailed investigations.

However, the conceptual reasoning behind the research aims and
hypotheses could be elaborated more convincing.

Self-report data is used. This is an important limitation.

The huge statistical power may also have made small and thus less
meaningful associations significant. Please discuss such potential bias
in more detail.

The practical implications could be illustrated with more detailed examples.

Author Response

In our Response to Reviewer #2

We deeply appreciate the reviewers for their thoughtful comments. We did our best to address all the comments/critics point-by-point, which are highlighted in yellow color. Three references are newly added, and they are listed on the last page.

Q1) However, the conceptual reasoning behind the research aims and hypotheses could be elaborated more convincing.

ANS1) Thanks for the comment. In our response to the comment, the aims and hypotheses are more elaborated as follows:

Anemia is a condition of not having enough healthy red blood cells to carry out oxygen to the body's tissues. In active skeletal muscles, the reduced oxygen-carrying ability of blood due to anemia interferes with cell oxygenation and weakens muscles [14]. As a result, anemia has been utilized as a surrogate marker for health problems. For example, previous studies have shown that anemia is associated with lower grip strength in older adults [15] and reduced grip strength and increased mortality in older inpatient patients [16].

Sarcopenia is a chronic condition characterized by losses in muscle mass and strength, and it is also considered a surrogate for a lower level of HRQoL, especially in elderly persons. From this perspective, loss of muscle strength assessed by handgrip strength is associated with health conditions, sarcopenia, disability, impaired physical functioning, and cognitive decline in older adults [17]. Additionally, handgrip strength is a biomarker for a lower HRQoL in older populations [18-20] and cancer survivors [21,22].

Taken together, it is reasonable to assume that anemia may predict a decline in HRQoL due to low handgrip strength. However, few studies have investigated if anemia and handgrip strength predict HRQoL later in life. In the current study, we aimed to investigate 1) the associations between anemia and handgrip strength with HRQoL and 2) the impact of anemia on HRQoL through weak handgrip strength in a representative sample of older Korean adults.

Q2) Self-report data is used. This is an important limitation.

ANS2) Thanks for the comments. Anemia, handgrip strength, and HRQoL were evaluated using the validated methods as follows:  “Serum hemoglobin was measured based on the SLS hemoglobin detection method with a Sysmex XN-9000 Automated Hematology Analyzer (Sysmex Corporation, Kobe, Japan), which uses cyanide-free sodium lauryl sulfate (SLS).” “Handgrip strength was measured with a digital hand dynamometer (digital grip strength dynamometer, T.K.K 5401, Takei Scientific Instruments Co., Ltd., Tokyo, Japan).” “HRQoL was assessed using the EuroQol-5 dimension (EQ-5D).”

However, the covariates were collected based on self-reported questionnaires. Therefore, an additional study limitation is added as follows: “Third, we cannot completely rule out the chance of measurement errors in the self-reported covariate questionnaire.”

Q3) The huge statistical power may also have made small and thus less meaningful associations significant. Please discuss such potential bias in more detail.

ANS3) Thanks for the comments. In our response to the comment, we understand that for an experimental study, too large a sample is unnecessary and unethical, and too small a sample is unscientific and also unethical. However, this is not an experimental study but a population-based study using nationwide survey data. In addition, we believe that on any hypothesis, scientific research is built upon determining the mean values of a given dataset. The larger the sample size, the more accurate the average values will be. Larger sample sizes also help researchers identify outliers in data and provide smaller margins of error or minimize bias. We would deeply appreciate the reviewer for his/her generosity and mercy in our reply.

Q4) The practical implications could be illustrated with more detailed examples.

ANS4) Thanks for the comment. In our response to the comment, the practical implications are elaborated in a more detailed manner as follows:

“Our study findings show that anemia may contribute to a lower HRQoL through weak handgrip strength, suggesting that promoting muscular strength via an active and healthy lifestyle is critical for better HRQoL in older Korean adults aged 65 years and older.”

List of Added References

  1. Puthucheary, Z.A.; Rawal, J.; McPhail, M.; Connolly, B.; Ratnayake, G.; Chan, P.; Hopkinson, N.S.; Phadke, R.; Dew, T.; Sidhu, P.S.; Velloso, C.; Seymour, J.; Agley, C.C.; Selby, A.; Limb, M.; Edwards, L.M.; Smith, K.; Rowlerson. A.; Rennie, M.J.; Moxham, J.; Harridge, S.D.; Hart. N.; Montgomery, H.E. Acute skeletal muscle wasting in critical illness. JAMA. 2013;310(15):1591-600. Erratum in: JAMA. 2014, 311, 625.
  2. Chang, S.Y.; Han, B.D.; Han, K.D.; Park, H.J.; Kang, S. Relation between handgrip strength and quality of life in patients with arthritis in Korea: the Korea National Health and Nutrition Examination Survey, 2015-2018. Medicina (Kaunas). 2022, 58, 172.
  3. Lee, S.H.; Kim, S.J.; Han, Y.; Ryu, Y.J.; Lee, J.H.; Chang, J.H. Hand grip strength and chronic obstructive pulmonary disease in Korea: an analysis in KNHANES VI. Int J Chron Obstruct Pulmon Dis. 2017, 12, 2313-2321.

Reviewer 3 Report

Thank you for inviting me to review this manuscript on “Relative Handgrip Strength Mediates the Relationship between Hemoglobin and Health-Related Quality of Life in Older Korean Adults.” I think the topic is very important for readers in the field of geriatric. There are several significant concerns in the study.

The number of people in the flowchart in Figure 1 does not match. Also, the number of participants in the abstract and the data source do not match the number of participants in Table 1.

Author Response

In our Response to Reviewer #3

We deeply appreciate the reviewers for their thoughtful comments. We did our best to address all the comments/critics point-by-point, which are highlighted in yellow color. Three references are newly added, and they are listed on the last page.

Q1) The number of people in the flowchart in Figure 1 does not match. Also, the number of participants in the abstract and the data source do not match the number of participants in Table 1.

ANS1) Thanks for the comments. We are sorry for this confusion, but we guess it happened by not explaining the flowchart in a detailed manner. Therefore, the description of the selection procedure of study participants including the flowchart is revised as follows:

For the current study purpose, we excluded 27,007 participants aged 19~64 years and included 10,484 older adults (4,532 men) aged 65 years and older only. Respondents with no information regarding HRQoL (n = 457), handgrip strength (n = 1557), health behaviors (n = 741), or covariates (n = 837) were excluded. Data from the remaining 6,892 individuals (3,139 males) were used in the final analyses.

Figure 1. Flowchart for the selection of study participants. KNHNES: Korea National

Health and Nutritional Examination Survey; HRQOL: health-related quality of life.

List of Added References

  1. Puthucheary, Z.A.; Rawal, J.; McPhail, M.; Connolly, B.; Ratnayake, G.; Chan, P.; Hopkinson, N.S.; Phadke, R.; Dew, T.; Sidhu, P.S.; Velloso, C.; Seymour, J.; Agley, C.C.; Selby, A.; Limb, M.; Edwards, L.M.; Smith, K.; Rowlerson. A.; Rennie, M.J.; Moxham, J.; Harridge, S.D.; Hart. N.; Montgomery, H.E. Acute skeletal muscle wasting in critical illness. JAMA. 2013;310(15):1591-600. Erratum in: JAMA. 2014, 311, 625.
  2. Chang, S.Y.; Han, B.D.; Han, K.D.; Park, H.J.; Kang, S. Relation between handgrip strength and quality of life in patients with arthritis in Korea: the Korea National Health and Nutrition Examination Survey, 2015-2018. Medicina (Kaunas). 2022, 58, 172.
  3. Lee, S.H.; Kim, S.J.; Han, Y.; Ryu, Y.J.; Lee, J.H.; Chang, J.H. Hand grip strength and chronic obstructive pulmonary disease in Korea: an analysis in KNHANES VI. Int J Chron Obstruct Pulmon Dis. 2017, 12, 2313-2321.

Round 2

Reviewer 2 Report

This manuscript related to the role of handgrip strength in mediating the relationship between Hemoglobin and Health-Related Quality of Life in Older is new. The study aimed to examine whether handgrip strength mediates the association between anemia and HRQoL. The results achieved are aligned with the objectives, and the results support the conclusions. All the questions asked were satisfactorily answered. 

I don’t have further questions on this paper.